# Switchable Ultra-Wideband All-Optical Quantum Dot Reflective Semiconductor Optical Amplifier

**DOI:** 10.3390/nano13040685

**Published:** 2023-02-10

**Authors:** Farshad Serat Nahaei, Ali Rostami, Hamit Mirtagioglu, Amir Maghoul, Ingve Simonsen

**Affiliations:** 1Photonics and Nanocrystals Research Lab. (PNRL), Faculty of Electrical and Computer Engineering, University of Tabriz, Tabriz 5166614761, Iran; 2SP-EPT Lab., ASEPE Company, Industrial Park of Advanced Technologies, Tabriz 5166614761, Iran; 3Nano and Quantum Technology, Stensgata 7c, 0358 Oslo, Norway; 4Department of Statistics, Faculty of Science and Literature, University of Bitlis Eren, Bitlis 13100, Turkey; 5Department of Physics, Norwegian University of Science and Technology (NTNU), 7491 Trondheim, Norway

**Keywords:** wideband optical amplifier, QDs, optical pumping, reflective semiconductor optical amplifier, selective amplification, filter

## Abstract

A comprehensive study has been conducted on ultra-broadband optically pumped quantum dot (QD) reflective semiconductor optical amplifiers (QD-RSOAs). Furthermore, little work has been done on broadband QD-RSOAs with an optical pump. About 1 μm optical bandwidth, spanning 800 nm up to 1800 nm, is supported for the suggested device by superimposing nine groups of QDs. It has been shown that the device can be engineered to amplify a selected window or a group of desired windows. Moreover, the operation of the device has been thoroughly investigated by solving the coupled differential rate and signal propagation equations. A numerical algorithm has been suggested to solve these equations. As far as we are concerned, a broadband optically pumped QD-RSOA that can operate as a filter has been introduced.

## 1. Introduction

Nowadays, quantum dot semiconductor optical amplifiers (QD-SOAs) have been a subject of interest because of their extensive applications in photonics, such as high-bit-rate optical switching, wavelength-division multiplexing, and signal processing [1,2,3]. There is a gap between the quantum dot (QD) levels and the wetting layer (WL). This gap makes them more promising than bulk and quantum well SOAs. In addition, it has been shown that QD-SOAs have lower noise figures and higher output power than bulk and quantum well amplifiers [4,5]. A comprehensive study has been conducted on the noise figures of these structures [5]. Furthermore, lower cross sections in carrier and photon interactions result in exciting features, such as lower gain saturation, lower frequency chirps, and shorter carrier relaxation lifetimes [6]. Current noises are usually seen at the device’s output in a broad frequency spectrum due to the high operation speed of SOAs. Hence, protection is vital for the electronic devices coupled to them against electromagnetic interferences, source instabilities, etc. [7]. Electrodes in electrically pumped QD-SOAs have uniform surfaces along the waveguide. Therefore, a constant injection current exists across the device’s length. However, the carrier concentration is not uniform along the SOA length due to slight variations in optical intensity. Furthermore, the injection current does not affect every point along the device uniformly. For example, the influence of the injection current on the noise figure is of more concern on the entrance of the device than its influence on the saturation power. On the contrary, when looking at the output facet, the impact of it on the saturation power is crucial [8]. Hence, intricate techniques have been suggested to adjust the injection current. Crosstalk can be reduced, gain linearity can be improved, and higher efficiency can be derived if optimized currents are utilized [8]. There are several reasons for using optical pumping instead of electrical pumping. First, fabrication is simplified and optical losses are minimized since we do not need to dope the device. Second, a uniform carrier distribution is achieved because there is no need to transport the generated carriers along the device. Third, a higher operation speed has been reported for optically pumped QD-SOAs [9]. Most importantly, integration is possible when we use an optical pump. Vertical cavity surface emitting lasers [10] and vertical cavity semiconductor optical amplifiers [11] are good examples of devices that utilize optical pumping.

Wavelength-division multiplexed passive optical networks are widely used due to the quick extension of internet traffic. These networks are commonly exploited in fiber-to-home setups for communication applications. Hence, several papers have been published to enhance optical communication features. In [12], the performance of multi-channel optical communication systems has been evaluated using digital nonlinearity compensation and electronic dispersion compensation. In [13], an approach has been suggested for QD-SOAs to generate signals with direct phase modulation. Optical amplification acquires more optical network bandwidth. Using reflective semiconductor optical amplifiers as colorless modulators in wavelength-separated optical networks has attracted significant attention recently [14,15]. In addition, wideband communication devices use these modern amplifiers [12,16]. These amplifiers have large modulation bandwidth, negligible crosstalk, and rapid dynamics in gain saturation [5,17,18,19]. However, they have a reflective facet on the rear part of the device, which further complicates their investigation due to counter-propagating signals in the cavity [20].

The solution process is a chemical process that is extensively used to synthesize quantum dots (QDs). In this method, the size of the QDs can be adjusted by the velocity of growth, the rotation speed of the solvent, the solvent’s concentration, and the temperature. Up to 0.1 nm size resolution has been reported for the QD’s radius [21,22,23,24,25,26].

Substantial progress in microfabrication and in our understanding of the nano-scale optical devices’ physics have resulted in the rapid growth of integrated photonics in the last two decades. This new field has been widely studied by researchers all around the world in recent years [27,28,29,30].

In this article, a comprehensive study has been done on QD-SOAs with optical pumping in their reflective configuration. Their dynamics of carrier densities, small-signal gain, and saturation attributes have been investigated. Unlike QD-SOAs with an electrical pump, an optical pump is needed in this device to insert carriers from the valence band to the corresponding conduction band. Hence, this pump can be integrated with the device on a single chip. In addition, they have proven to have better gain recovery and higher operation speed [9]. An ultra-broadband optical bandwidth has been achieved by superimposing QDs with different radii, utilizing solution process technology that guarantees up to 0.1 nm size variation for the QDs. In addition, an approach has been suggested to make selective amplification possible. Three windows have been selected to show the mentioned feature. These windows have been chosen due to their extensive application in optical communication. In addition, the gain profile of the device has been plotted versus the input signal power for two distinct pump powers. QD-RSOAs are used in the next-generation wavelength-division multiplexed (WDM) optical networks. In a nutshell, an ultra-broadband, optically pumped quantum dot reflective semiconductor optical amplifier has been engineered in this paper for the first time, which is also capable of operating as a filter. This device is optically pumped, making it easier to be integrated with other optical devices.

## 2. Concept and Modeling

### 2.1. Proposed Structure

Three groups of QDs have been exploited to implement our model, as depicted in Figure 1. A mirror has been used at the rear facet to reflect the traveling signals and bring about the double-pass mechanism. The QDs are made from InGaAs, with AlAs acting as cladding layers. The pump and signals can be coupled from the right side to the proposed structure. The materials are selected in a way that they operate in the desired window. After choosing the materials, the model was simulated by COMSOL Multiphysics to identify the eigenenergies. An array of 3 × 3 QDs was simulated with periodic boundary conditions at both sides. In addition, the QD’s radius has been swept. Figure 2 illustrates the emission energy as a function of the radius of the QDs. One can choose the appropriate radius based on the desired wavelength. Here, we have selected R1, R2, and R3 to cover these three wavelengths (1.55 µm, 1.5 µm, 1.31 µm).

Further simulations uncovered that there are two confined energies for all three wavelengths, as seen from the results presented in Figure 3, Figure 4 and Figure 5. Although there seem to be more than two confined energy states, they are apart by less than 25 mV. As a result, these adjacent energy states are modeled as one. In addition, they are unbound energy states close to each other. They are modeled as a thick wetting layer state in the proposed structure.

Figure 6 depicts a schematic of the QDs. As mentioned, we have two confined energy states called the Ground State (GS) and the Excited State (ES), respectively. A set of energy states outside are denoted by a Wetting Layer (WL). There are several transitions from these states that are taken into account in the rate equations. The electrons can move from the WL to the ES, from the ES to the GS, and vice versa. Electrons can also recombine radiatively from the GS due to stimulated emission. This phenomenon is what is causing the amplification of our signal. In other words, we have got a signal that provokes the electrons in the GS to recombine. As a result, we have two signals with the same direction and amplitude at the output. These signals add up constructively, making amplification possible. The carriers are injected with a pump from the valence band. This pump must provide E_p_ to pump the carriers to the ES in the conduction band. Otherwise, we will not have enough carriers, and the system will not work.

### 2.2. Selective Amplification

As mentioned previously, three classes of QDs have been chosen in this paper. The energy E_p_ must be adjusted for each of these groups to be activated. Figure 7 depicts the energy difference between the ESs in the conduction and the valence band versus the QD’s radius. We know that without enough carriers in the conduction band, the QDs cannot perform their task as an amplifier. This fact is used to have selective amplification for any desired group with its specified radius. In other words, by applying a pump with the energy of E_p1_, only the QDs with the corresponding radii R1 are activated, and the other two groups are deactivated. In addition, pumps with the energy E_p2_ and E_p3_ can activate QDs groups with radii R2 and R3, respectively.

### 2.3. Homogenous and Inhomogeneous Broadenings

In the presented model, three groups of QDs have been selected. We divide each group into n sub-groups. Each mode has an energy separation of ∆*E* calculated by:(1)$$\Delta E=\frac{ch}{2nL}$$
where *h* and *c* are the Planck constant and velocity of light in a vacuum, respectively. The cavity’s length is denoted by *L* and the refractive index by *n*. Energies in each group can be expressed as:(2)$$E_{i,n}=E_{i}-\left(M-n\right)\Delta E\;\;\;\;\;\;\;\;\;\;\;\;\;\;\;i=1,2,3$$

In this case, the main modes in each group are represented by *E*_1_ = *E*_1,*n*_, *E*_2_ = *E*_2,*n*_, and *E*_3_ = *E*_3,*n*_. Homogenous Broadening (HB) and Inhomogeneous Broadening (IHB) can be derived by [2]:(3)$$G_{i}=\frac{1}{\sqrt{2\pi \xi^{2}}}\;\exp\left[\frac{-{\left(E_{i,n}-E_{i}\right)}^{2}}{2\xi^{2}}\right]$$
(4)$$B_{i,mn}\left(E_{i,m}-E_{i,n}\right)=\frac{\Gamma_{h}/2\pi }{{\left(E_{i,m}-E_{i,n}\right)}^{2}+{\left(\Gamma_{h}/2\pi \right)}^{2}}$$
where Γ*_h_* denotes the Full Width Half Maximum (FWHM) of the Lorentzian profile, and *ξ* the coverage of the channel. If we use N groups of QDs with distinct radii, the total Gaussian profile can be obtained by:(5)$$G_{total}=\overset{N}{\underset{i=1}{\sum }}G_{i}$$

Figure 8 represents the IHB and HB for the three specified groups.

### 2.4. Numerical Algorithm

There are two sets of coupled differential equations that must be solved to study the operation of the QD-RSOAs. They will be introduced comprehensively in the following sections. Several methods have been proposed to solve this problem [31,32,33]. In this article, signal powers and occupation probabilities are evaluated at each point using the time–space discrete technique. Equations are introduced in a definite time frame rather than retarded time frame [1,34]. First, the equations are obtained in the retarded time frame. Then the ultimate answers are extracted in the absolute time frame. In a nutshell, after solving the problem, the concluding answers are shifted. The influence of counter-propagation signals is taken into account by utilizing this approach. To quantify spatial and time derivatives, the Euler approximations are used [33]. Let ∆t denote the time steps in the proposed algorithm. Hence, for our answers to converge, the spatial steps are evaluated by V_g_∆t. To figure out the optimum step size, several quantities were assessed. Finally, we derived enough accuracy and stability by implementing ∆t = 50 fs. Furthermore, at each spatial step, pulses are delayed by ∆t. To derive the dynamic results, static simulation outcomes are utilized [35,36]. Figure 9 demonstrates how the solutions are extracted using this numerical approach.

Step 1: the input signal enters at y = 0 after attaining the steady-state condition. A Gaussian time profile has been assigned for this input signal.

Step 2: pump power, signal power, and carrier densities are extracted at the subsequent spatial step.

Step 3: we propagate the forward signals through the cavity. Meanwhile, the effect of the backward signals is not taken into account at this step. After reaching the rear side of the cavity, the signals are either absorbed or reflected by the mirror. The mirror’s specifications define the proportion of the reflected signal. In the ideal case, no absorption is seen, and all the signal is reflected by the mirror (R = 1). However, in reality, we cannot attain this goal.

Step 4: we update the backward signals while keeping the forward signals constant. In this step, we make sure that this process is repeated until convergence has been reached and an acceptable tolerance is achieved.

Figure 10 demonstrates the proposed flowchart to implement the described procedure above. By using this method, we also automatically consider the effect of the counter-propagating pulses in the interaction points. Hence, the results would be more accurate.

## 3. Results and Discussion

In this section, the rate and signal propagation equations are first introduced. To solve these coupled differential equations, we have used the mentioned numerical algorithm. In addition, the equations are solved numerically in MATLAB. It has been shown that the suggested model has an ultra-wideband optical gain with switching capability. In the end, the operation of the device, in pulse and CW operation mode, has been studied.

### 3.1. Coupled Differential Rate and Signal Propagation Equations

As mentioned, there are two confined energy states in the valence and the conduction bands. In addition, a thick wetting layer is considered in the proposed model. Rate equations can specify the optical properties of the mentioned QD-RSOA [1,37,38]. However, we must include the effect of the rear facet in the suggested equations. The mirror on this facet reflects the forward signals. The following equations point out the dynamics of the allowed energy states.
(6)$$\frac{\partial N_{W}(y,t)}{\partial t}=\frac{N_{Q}h_{n,i}}{\tau_{e-w}}-\frac{N_{W}(1-h_{n,i})}{\tau_{w-e}}-\frac{N_{W}}{\tau_{w-r}}$$
(7)$$\frac{\partial h_{n,i}\left(y,t\right)}{\partial t}=\frac{f_{n,i}\left(1-h_{n,i}\right)}{\tau_{g-e}}+\frac{N_{W}(1-h_{n,i})}{N_{Q}\tau_{w-e}}-\frac{h_{n,i}\left(1-f_{n,i}\right)}{\tau_{e-g}}-\frac{{h_{n,i}}^{2}}{\tau_{e-r}}-\frac{h_{n,i}}{\tau_{e-w}}+\frac{v_{g}\alpha_{\max}}{N_{Q}}(1-2h_{n,i})(P_{P}{}^{+}(y,t)+P_{P}{}^{-}(y,t))$$
(8)$$\frac{\partial f_{n,i}(y,t)}{\partial t}=-\frac{f_{n,i}(1-h_{n,i})}{\tau_{g-e}}+\frac{h_{n,i}(1-f_{n,i})}{\tau_{e-g}}-\sum_{m}^{}\frac{v_{g}g_{i,mn}^{}}{N_{Q}}(2f_{n,i}-1)(P^{+}(y,t)+P^{-}(y,t))-\frac{f_{n,i}{}^{2}}{\tau_{g-r}}$$

Here electron occupation probabilities in the GS and the ES are represented by *f* and *h*, respectively. The electron charge is denoted by *q*. The electron density in the WL is also symbolized by *N_w_*. *τ_w−e_* and *τ_e−w_* are the electron relaxation lifetime from the wetting layer to the excited state and the electron escape lifetime from the excited state to the wetting layer, respectively. *τ_w−r_*, *τ_g−r_*, and *τ_e−r_* are the spontaneous radiative lifetimes in the WL, GS, and ES. In addition, the electron relaxation lifetime from the ES to the GS is denoted by *τ_e−g_*. The electron escape lifetime from the GS to the ES is shown by *τ_g−e_*. *N_Q_* and *V_g_* are the volume density of the QDs, and the group velocity of light, respectively. *α*_max_ identifies the maximum modal absorption coefficient. As can be seen, the term (1 − 2*h*) defines ES carrier dynamics. When 2*h* > 1, we have optical gain. However, when it is less than one, absorption is the dominant mechanism. Due to the optical pump power (last term in Equation (7)), h escalates toward 0.5. As a result, the term 1 − 2*h* vanishes. In this case, a transient decrease in the h value makes the term 1 − 2*h* positive. Hence, absorption is achieved using optical pumping [39]. *g_i,mn_* denotes the linear gain for a specified group of QDs (n) and a particular photon mode (m) [19,40].
(9)$$g_{i,mn}=\frac{h\pi q^{2}}{c\varepsilon_{0}n_{r}m_{0}^{2}E_{i,n}}\langle \left|\hat{e}.p_{cv}^{2}\right|\rangle {\left|M_{env}\right|}^{2}N_{Q}G_{i}DB_{cv}(E_{i,m}-E_{i,n})$$
where the free electron mass and charge are denoted by *m*_0_ and *q*, respectively. *ε*_0_ is the vacuum permittivity. The refractive index is denoted by *n_r_*. *D* is the degeneracy rate, and *N_Q_* is the QD volume density. |*M_env_*|^2^ and <|*e.p_cv_*|^2^> are the envelope function and the momentum matrix element.

Pump and signal propagation equations are also solved concurrently with the rate equations to study the performance of the proposed model [1]. The forward and backward signals are distinguished by a negative sign.
(10)$$\frac{\partial P^{+}(y,t)}{\partial y}+\frac{1}{v_{g}}\frac{\partial P^{+}(y,t)}{\partial t}=(\Gamma g_{w_{s}}(y,t)-\alpha_{\mathrm{int}})P^{+}(y,t)$$
(11)$$-\frac{\partial P^{-}(y,t)}{\partial y}+\frac{1}{v_{g}}\frac{\partial P^{-}(y,t)}{\partial t}=(\Gamma g_{w_{s}}(y,t)-\alpha_{\mathrm{int}})P^{-}(y,t)$$
(12)$$\frac{\partial P_{P}{}^{+}(y,t)}{\partial y}+\frac{1}{v_{g}}\frac{\partial P_{P}{}^{+}(y,t)}{\partial t}=(-\alpha_{\max}(1-2h)-\alpha_{\mathrm{int}})P_{P}{}^{+}(y,t)$$
(13)$$-\frac{\partial P_{P}{}^{-}(y,t)}{\partial y}+\frac{1}{v_{g}}\frac{\partial P_{P}{}^{-}(y,t)}{\partial t}=(-\alpha_{\max}(1-2h)-\alpha_{\mathrm{int}})P_{P}{}^{-}(y,t)$$
where *P^+^*, *P^−^*, *P_p_^+^*, and *P_p_^−^* represent the backward and forward signal and pump powers. *α*_int_ denotes the material absorption coefficient. The modal absorption coefficient of the pump can be derived from *α*_abs_ = −*α*_max_(1 − 2*h*) − *α*_int_. Material gain is denoted by $g_{w_{s}}$, which can be calculated by:(14)$$g_{w_{s}}(y,t)=\sum_{n=1}^{2M+1}g_{mn}(2f_{n}-1)$$

### 3.2. Stimulation Parameters

Table 1 represents the parameters that are utilized to derive our results [2,18,41,42,43].

### 3.3. Ultra-Wideband Optical Gain

As discussed before, we achieve a Gaussian profile for each group of the QDs with a corresponding radius in the energy profile. Hence, using more QDs with distinct radii, their effects are superimposed. In addition, each QD group has a unique center wavelength, around which we have a Gaussian distribution. As a result, an ultra-broadband optical gain can be achieved. Figure 11 represents the optical gain for the seven and nine groups of QDs. As can be seen, we have amplification for photons with the energy from 0.7 eV up to 1.5 eV. Using the Equation E = hc/λ, one can calculate the wavelength spectrum for which there is an amplification. Furthermore, by increasing the QD groups, higher optical gain is achieved. This is evident since, by increasing QD groups, N_Q_ is increased. This leads to higher optical gain. This is why the optical gain is about 32 dB for nine groups of QDs; whereas this number is about 18 dB for seven groups. Furthermore, a smoother surface is achieved by using more QD groups. However, increasing the QDs results in a more sophisticated model, therefore increasing the cost of fabrication. Hence, there is a trade-off between the device’s performance and the cost of fabrication. One can engineer the desired model based on one’s needs, using the appropriate number of groups with distinct radii. In addition, one can alter the materials of the QDs and the cladding layers to cover a different wavelength spectrum. The only thing that must be taken care of is the procedure, which is described in the concept and modeling section. First, the energy levels for the corresponding materials must be calculated. Then, by sweeping the QD’s radius, energy differences between the conduction and valence band for each eigenstate versus the QD’s radius must be derived. Figure 2 and Figure 6 depict these profiles for QD mode from InGaAs with AlAs claddings.

### 3.4. Selective Amplification

In the previous section, an ultra-broadband optical gain was achieved by using nine and seven groups of QDs. In this section, it will be shown that not only is this device capable of amplifying a large optical bandwidth signal, but also that this amplification is selective. In other words, we can activate each group individually or simultaneously. When all the groups are activated simultaneously, amplification is possible for all the wavelength spectrums. However, if a group is activated, only the spectra corresponding to that group are amplified. This is very useful in wavelength-division multiplexing. The corresponding pump power for each channel can activate that. As mentioned before, amplification is possible in the proposed model due to stimulated emission. However, to have amplification, one must satisfy the population inversion condition. This means that carriers must be inserted into the device’s conduction band, either by an injection current or an optical pump. Optical pumping satisfies the population inversion condition in this paper. In other words, carriers are pumped from the excited state in the valence band to the excited state in the conduction band. To implement this approach, the pump power must be adjusted to provide the energy needed for this action. This means that even if we have nine groups of QDs with distinct radii, amplification is only applicable for those that have the corresponding pumping. For example, here we have selected three optical windows (1550 nm, 1500 nm, and 1310 nm). These windows have been selected due to their significance in communication applications. Afterward, the required QDs’ radii can be derived. The required radiation energy for these windows can be calculated using E = hc/λ (0.7999 eV, 0.8266 eV, 0.9464 eV). Then the required radii to emit these energies are found (R1, R2, R3). In the last step, the required pump energy to activate each window is derived. Figure 12, Figure 13 and Figure 14 illustrate the optical gain when each of these channels is activated individually. As can be seen, amplification has just been applied to one channel, and the other two do not experience it. The first channel (1550 nm) has been amplified by about 7.5 dB. However, the second channel (1500 nm) and the third channel (1310 nm) experience 8 dB and 9.8 dB of amplification, respectively. So we have a slight variation in the amplification for each channel. This is because the radius for each QD varies slightly. In our simulations, we have assumed that the number of QDs is selected in a way that they have no interaction with each other in the specified volume. As a result, N_Q_ (volume density of QDs) is distinct for each of these channels, which explains the difference in the amplification quantities between them.

### 3.5. Gain Versus Length

In the previous sections, the wideband optical gain and selective amplifications for three channels were demonstrated. In the following parts, we will investigate the operation of the device for a single optical window (1500 nm). This is done to decrease the stimulation time due to different parameters’ sweep. First, the effect of the device’s length on the gain value has been studied in Figure 15. The length of the device has been varied and results are presented. As depicted, the gain value rises with length up to a certain point (L_max_ ≈ 6 mm). After this length, the device saturates, and the gain drops. This is because, in this region, the gain is dominated by material loss. As a result, the device attenuates the signal instead of amplifying it. In this paper, a 2 mm length has been selected for the amplifier. Hence, it is guaranteed that we are not in the saturation regime.

### 3.6. Dynamic Characteristics

When an input pulse is inserted into the device, stimulated emission provokes the carriers in the GS to be recombined radiatively. The GS is refilled by the carriers in the ES. In other words, the ES performs like carrier storage, and therefore, it needs to be refilled constantly. The ES can be refilled by carrier absorption via suitable optical pumping. In addition, relaxed carriers from the wetting layer minorly contribute to this process. These carriers are recaptured from the ES to the WL. The relaxation lifetime constant is used to denote the rate of this process. The electron occupation probabilities in the GS and the ES versus time are depicted in Figure 16 for two distinct optical pumping powers (80 mW, and 200 mW). As can be seen, the recovery time is lower for the higher optical pumping power. As a result, f and h recover faster when a 200 mW optical pumping power is utilized. This is because more carriers are absorbed for the higher optical pumping power. In other words, the excited state is refilled faster in this situation. The dynamic characteristics of the electron concentration in the WL (Nw) for distinct pumping powers are presented in Figure 17, demonstrating a faster dynamic for 200 mW optical pumping power. This is because the carriers move from the ES to the WL with a lifetime constant equal to τ_ew_. Since more carriers are absorbed to the ES in the case of higher optical pumping power, more of them move to the wetting layer with the time rate of τ_ew_. Figure 18 depicts the time evolution of f and h for different input powers. Higher input powers engage more carriers in the amplification process. As a result, the transient drops in f and h values are higher when high input power is utilized. This can be somehow recompensated by using higher optical pump power. As a result, more carriers are absorbed. Hence, the decrease in f and h values will be lower.

### 3.7. Gain Versus Input Power

The gain value versus input power for two distinct pump powers has been illustrated in Figure 19. This figure specifies the saturation power. For example, the saturation power is about 3 dBm in the case of 10 mW input power. Several parameters control the saturation power. For instance, the number of quantum dot layers and the electron capture rate of the quantum dots are two crucial factors. In addition, increasing the QD layers enhances the modal gain. However, a higher optical pump is required to obtain an identical output saturation power. This is because, by doing so, one can get a similar inversion level. Furthermore, the minimum relaxation rate of the electrons from the excited state to the ground state restricts the maximum saturation power. The quasi-equilibrium between the quantum dots and the reservoir electron states falls apart in this region. Therefore, intense spectral hole burning occurs. In a nutshell, the maximum saturation power and the capture time are inversely related to each other [5].

### 3.8. Optical Bandwidth Comparison

Table 2 represents some reported optical amplifiers and their corresponding optical bandwidth and operation regime. As can be seen, about 1 μm optical bandwidth has been achieved in this paper, which is a significant improvement compared to the others.

## 4. Conclusions

The design of a novel all-optical ultra-broadband reflective semiconductor optical amplifier has been proposed. The wideband capability of the device is demonstrated for seven and nine groups of QDs. In addition, an approach has been suggested to personalize the desired amplification windows by engineering the QDs’ radii and the optical pump. These windows can be activated simultaneously or one at a time. They are activated by applying appropriate optical pumping, which is discussed comprehensively in Section 2.1 and Section 3.4. Three windows (1500 nm, 1550 nm, and 1310 nm) have been selected as an example to prove the switching capability of the proposed device. These windows have been selected due to their significance in optical communication and wavelength-division multiplexing. The gain saturation characteristics in the CW operation point are discussed, and the maximum length for the device has been determined. Rate and signal propagation equations have been solved simultaneously to study the structure. They have coupled differential equations, and analytical solutions for them do not exist. An algorithm has been suggested to solve these equations numerically. In addition, the pulse operation mode has been studied. Time dependency of the ground state and excited state and their recovery time has been discussed. In the end, the saturation power for the device has been introduced.

In this paper, InGaAs quantum dots with AlAs claddings have been utilized. Numerical simulations demonstrate that these quantum dots can bring about amplification for the O, E, S, C, and L bands. Regardless, future research could continue to examine different materials and repeat the same procedures in this paper to derive amplification for another spectrum of frequencies. The proposed structure performs as an all-optical device. Hence, it is highly practical and easy to integrate with other optical devices. In addition, the ultra-broadband operation and switching capability of this device make it a good candidate for wavelength-division multiplexing in passive optical networks. Integration of the device with other optical devices would undoubtedly result in exciting applications. By introducing the suggested model, one step is taken toward all-optical integrated circuits.

## Figures and Tables

**Figure 1 nanomaterials-13-00685-f001:**
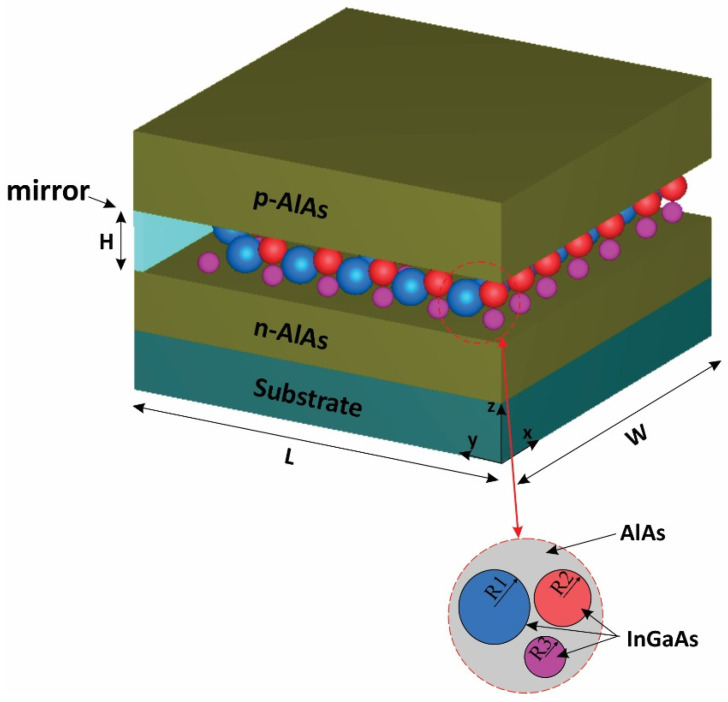
The proposed structure for three groups of quantum dots.

**Figure 2 nanomaterials-13-00685-f002:**
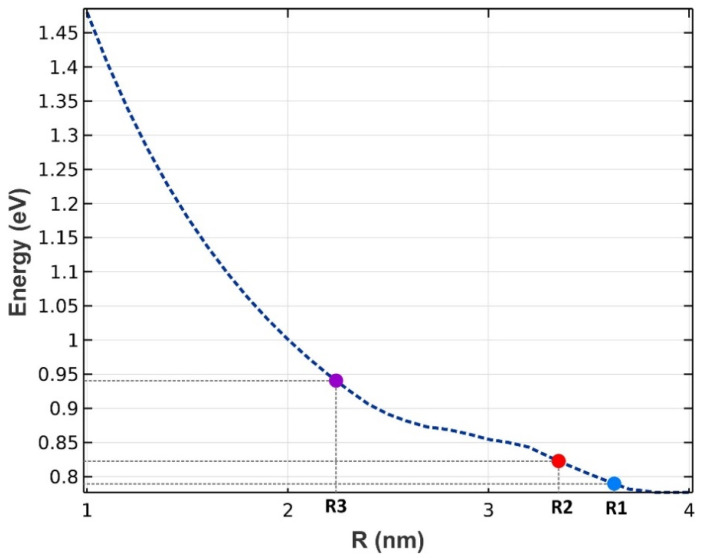
The energy difference between the ground states in the conduction band and the valence band (radiated energy) as a function of the QD’s radius. Three groups have been chosen (R1, R2, and R3).

**Figure 3 nanomaterials-13-00685-f003:**
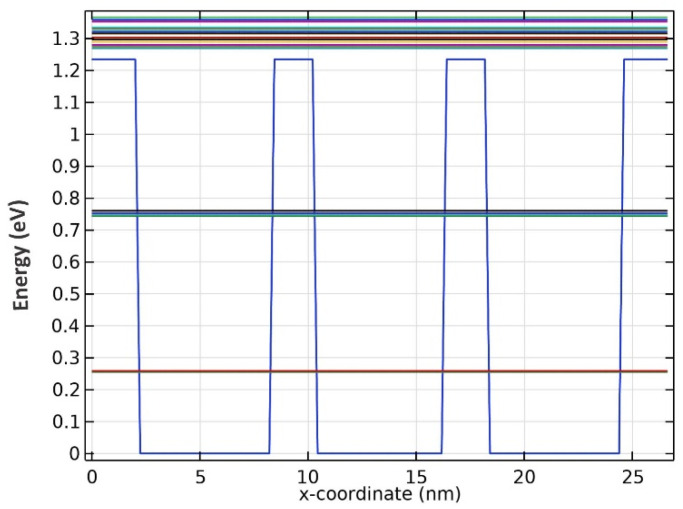
Energy band diagram for QDs with R1 radii.

**Figure 4 nanomaterials-13-00685-f004:**
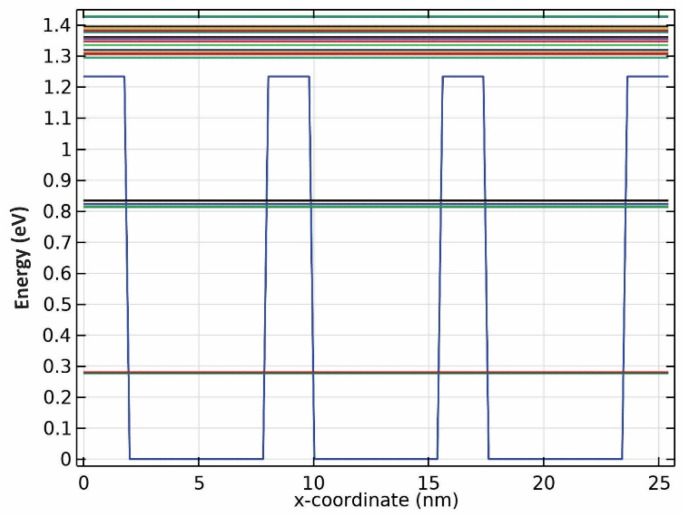
Energy band diagram for QDs with R2 radii.

**Figure 5 nanomaterials-13-00685-f005:**
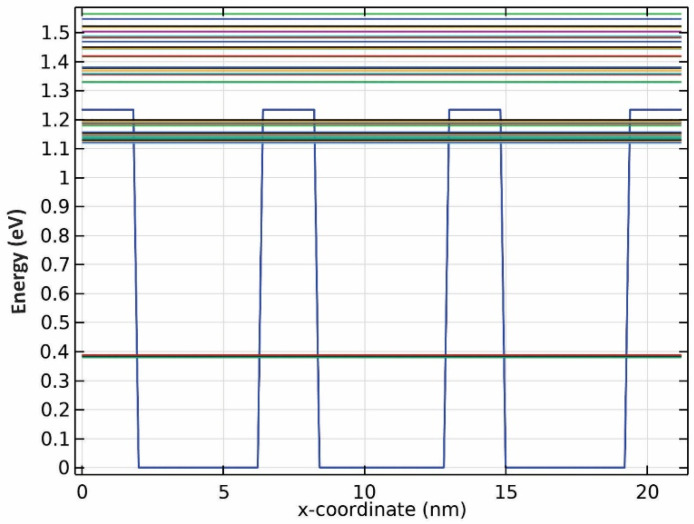
Energy band diagram for QDs with R3 radii.

**Figure 6 nanomaterials-13-00685-f006:**
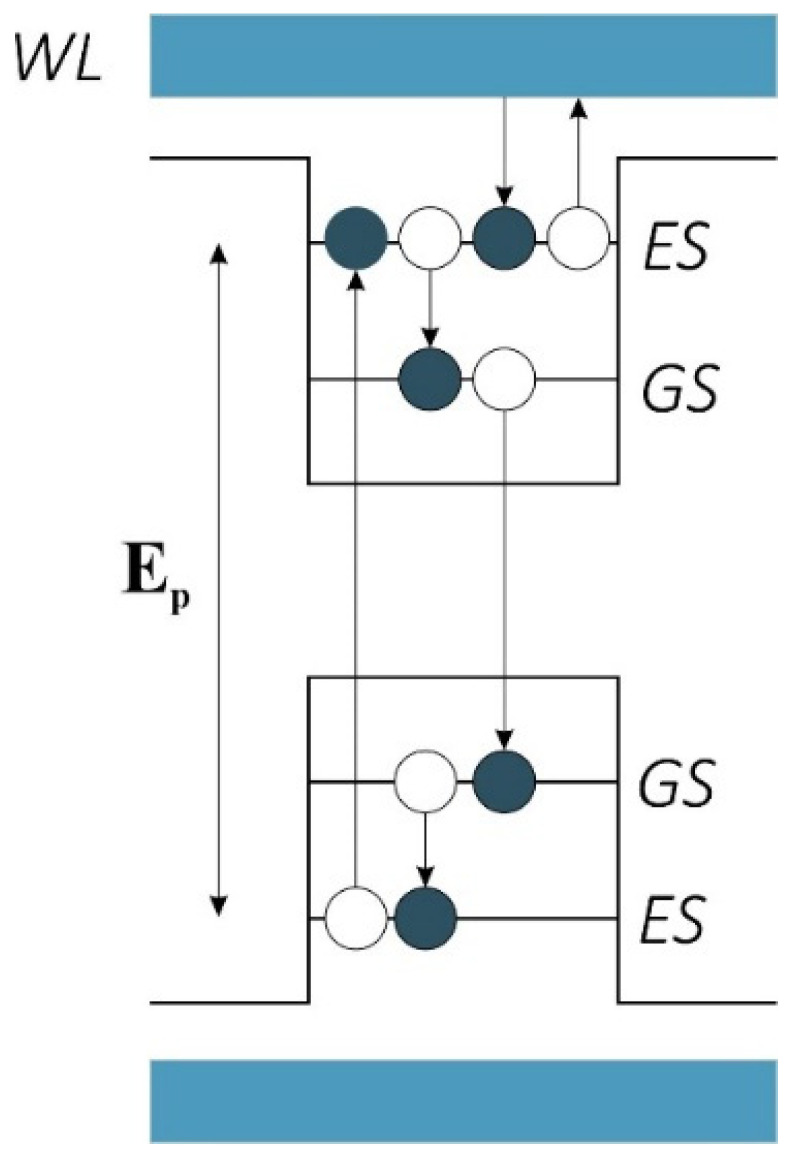
Transitions between the eigenstates.

**Figure 7 nanomaterials-13-00685-f007:**
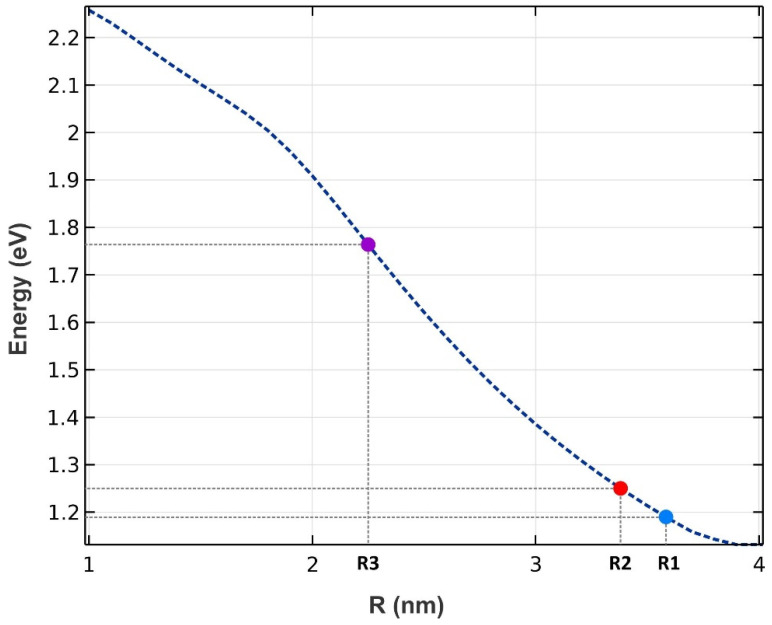
The energy difference between the excited states in the conduction band and the valence band (pump energies) as a function of the radii of the QDs. Three groups of QDs have been chosen (R1, R2, and R3).

**Figure 8 nanomaterials-13-00685-f008:**
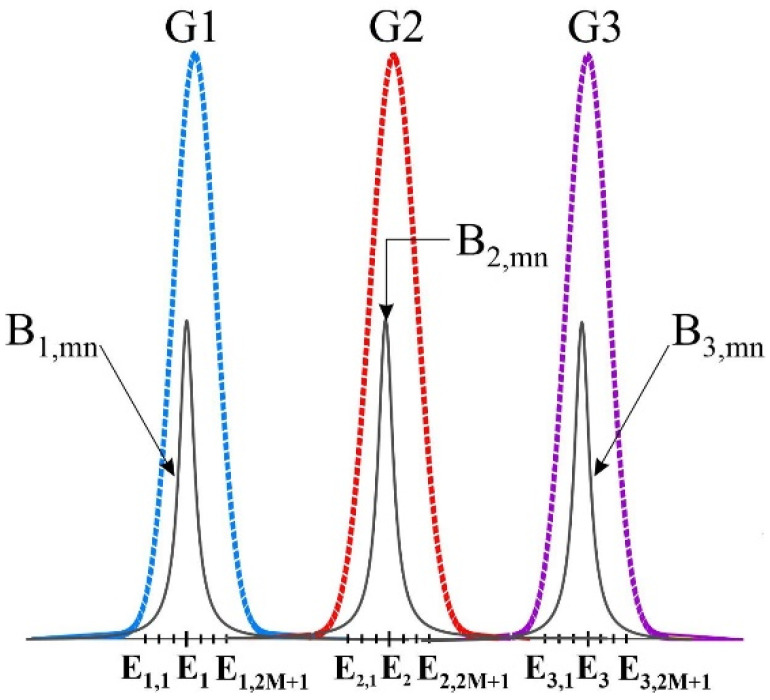
Homogenous and Inhomogeneous Broadenings for three groups of QDs.

**Figure 9 nanomaterials-13-00685-f009:**
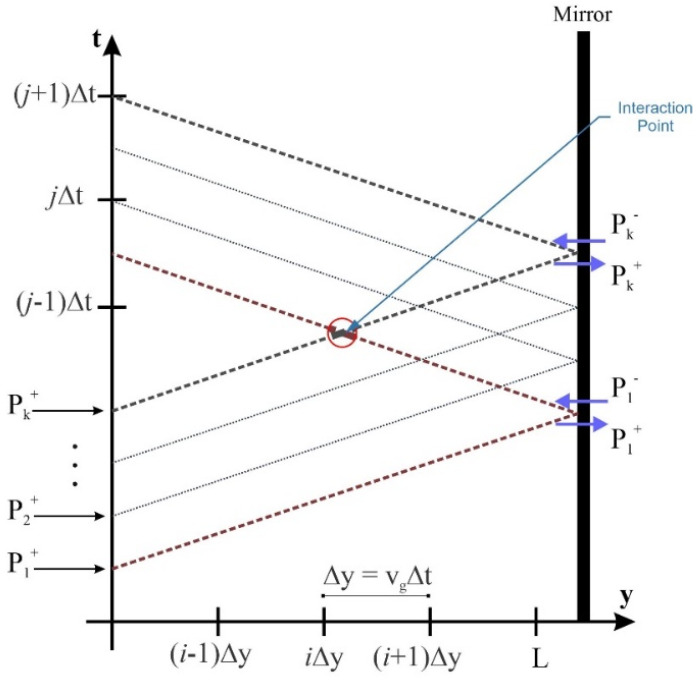
Forward and backward signals in discrete time and space matrix.

**Figure 10 nanomaterials-13-00685-f010:**
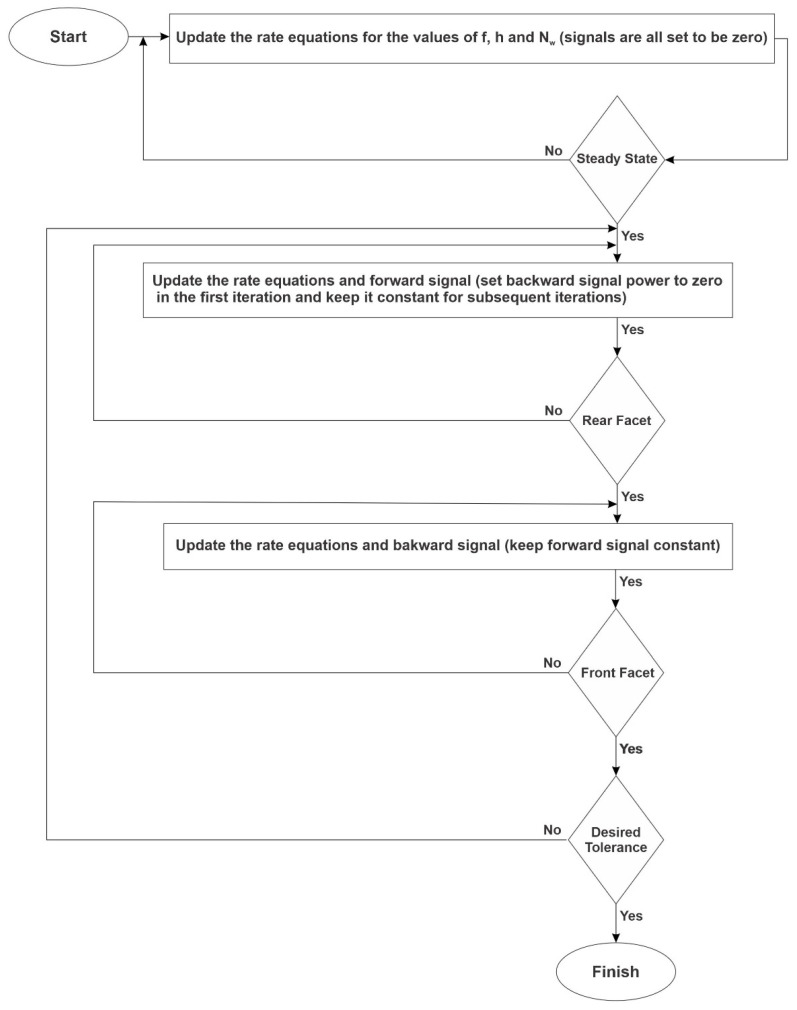
Flowchart for the proposed algorithm.

**Figure 11 nanomaterials-13-00685-f011:**
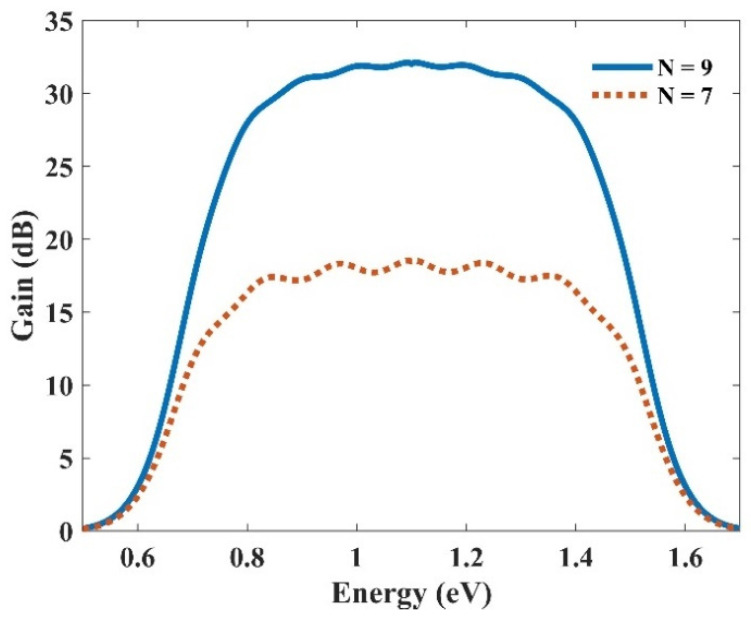
Ultra-wideband optical gain, obtained by superimposing seven and nine groups of QDs.

**Figure 12 nanomaterials-13-00685-f012:**
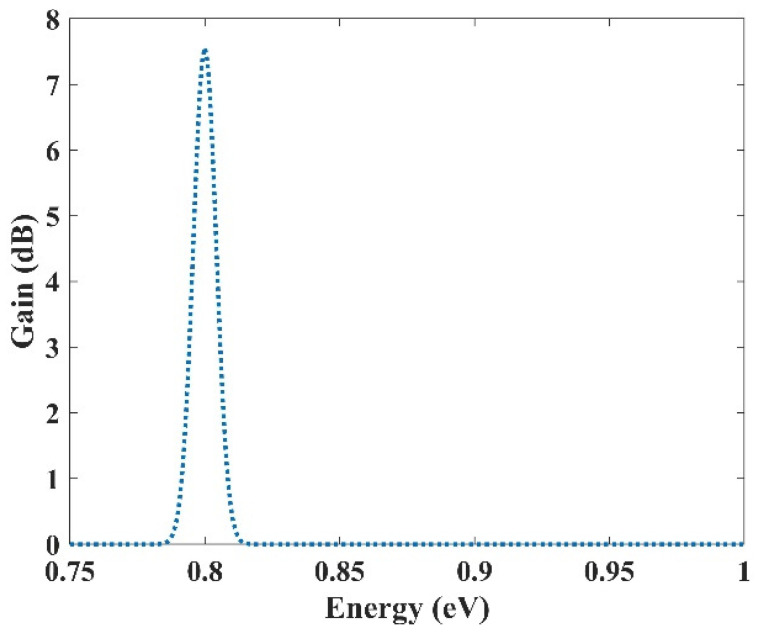
Selective amplification for channel 1 (1550 nm).

**Figure 13 nanomaterials-13-00685-f013:**
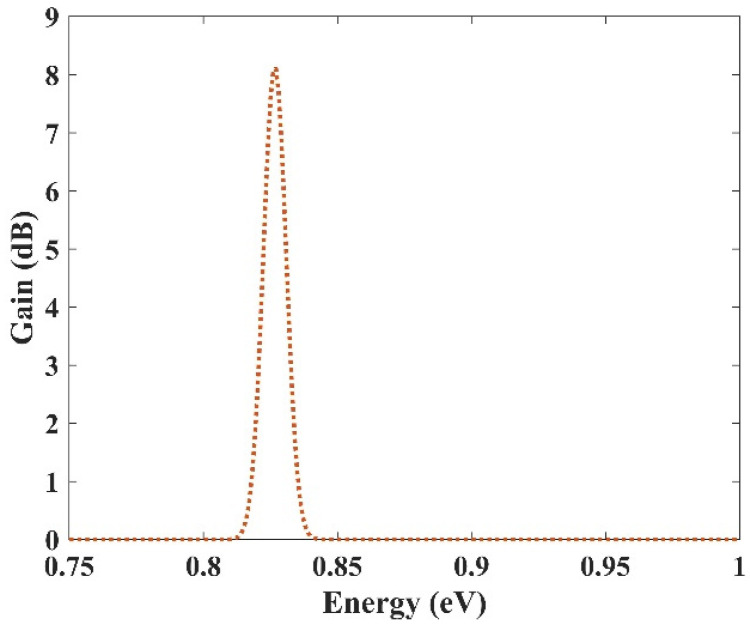
Selective amplification for channel 2 (1500 nm).

**Figure 14 nanomaterials-13-00685-f014:**
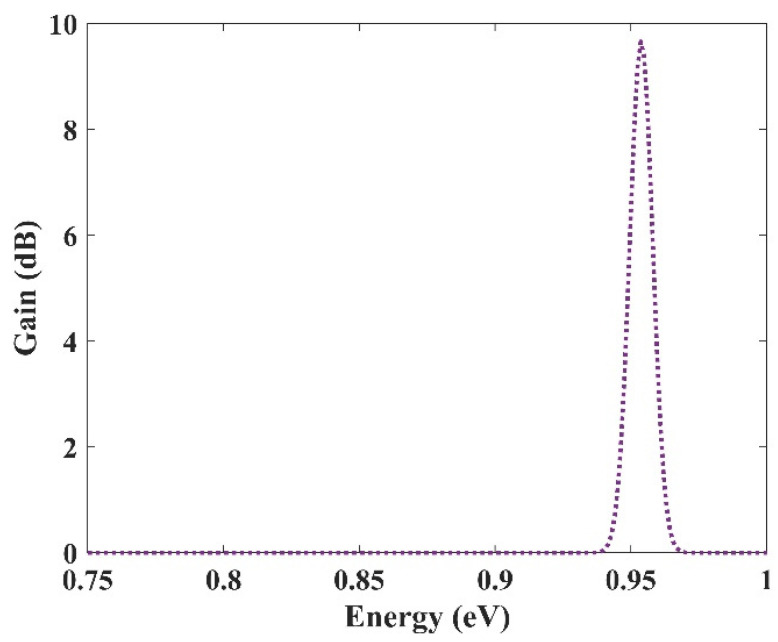
Selective amplification for channel 3 (1310 nm).

**Figure 15 nanomaterials-13-00685-f015:**
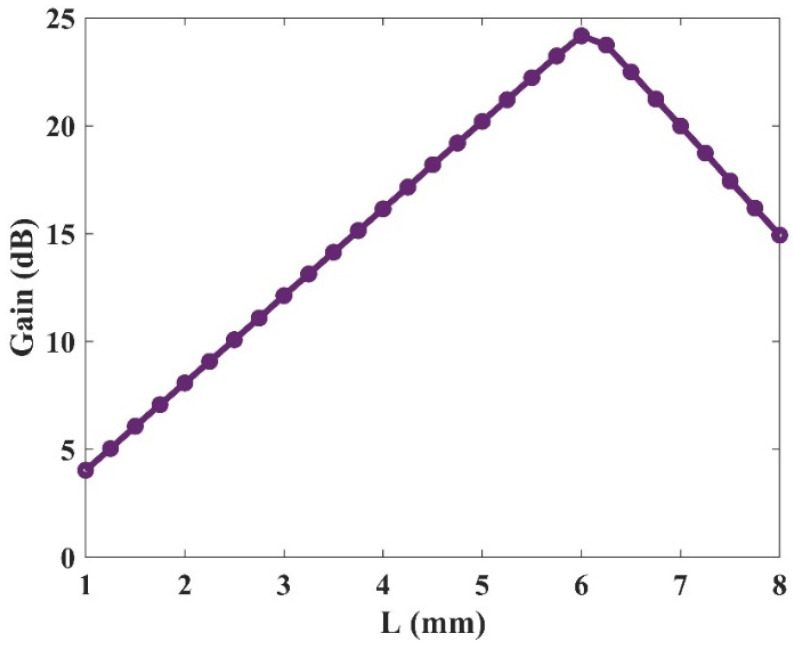
Optical gain versus the QD-RSOA’s length with 80 mW optical pumping.

**Figure 16 nanomaterials-13-00685-f016:**
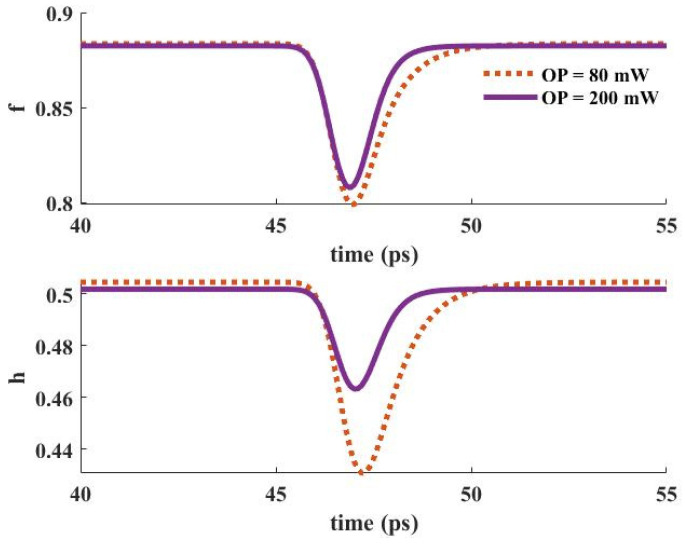
Occupation probability in the ground and excited states versus time for distinct pump powers (input signal power = 1 μW). f and h can vary between 0 to 1.

**Figure 17 nanomaterials-13-00685-f017:**
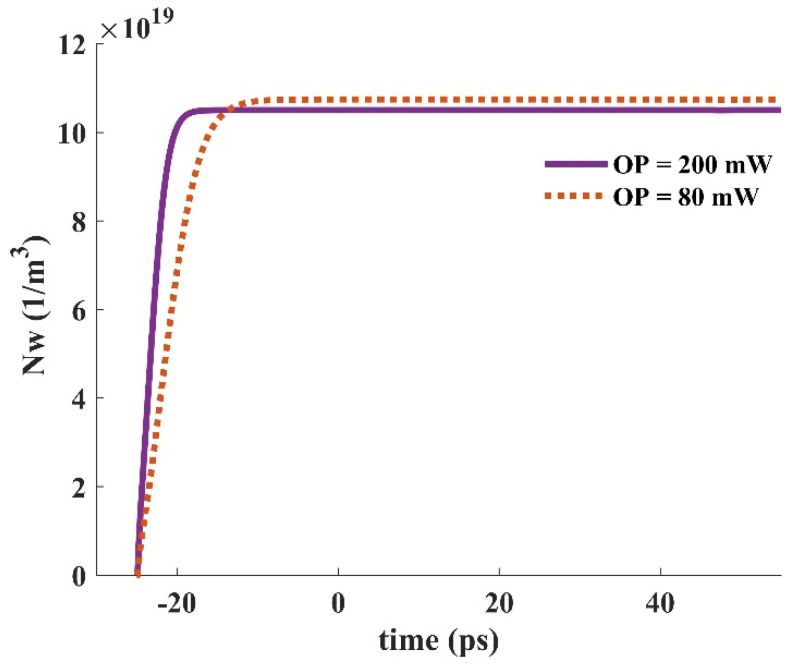
The electron concentration in the wetting layer for distinct optical pump power (input signal power = 1 μW).

**Figure 18 nanomaterials-13-00685-f018:**
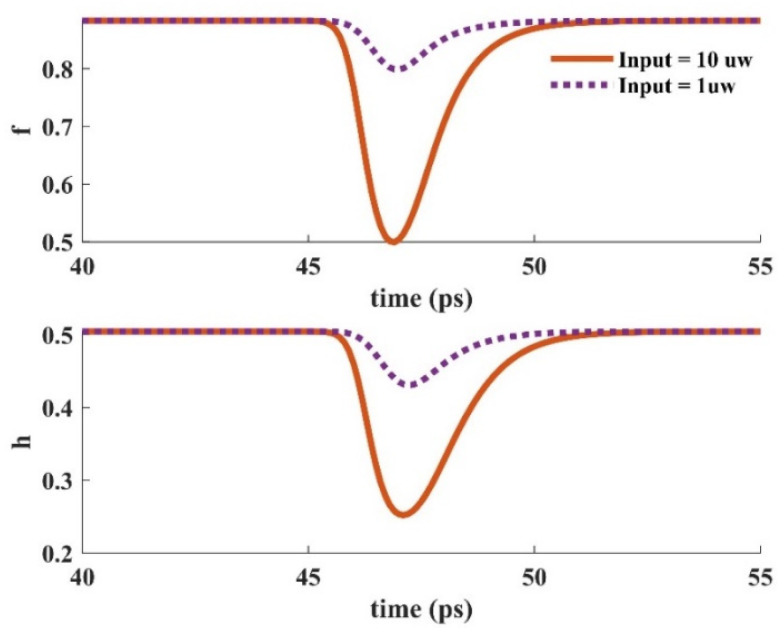
Occupation probability in the ground and excited states versus time for distinct input signal powers (Pump power = 80 mW).

**Figure 19 nanomaterials-13-00685-f019:**
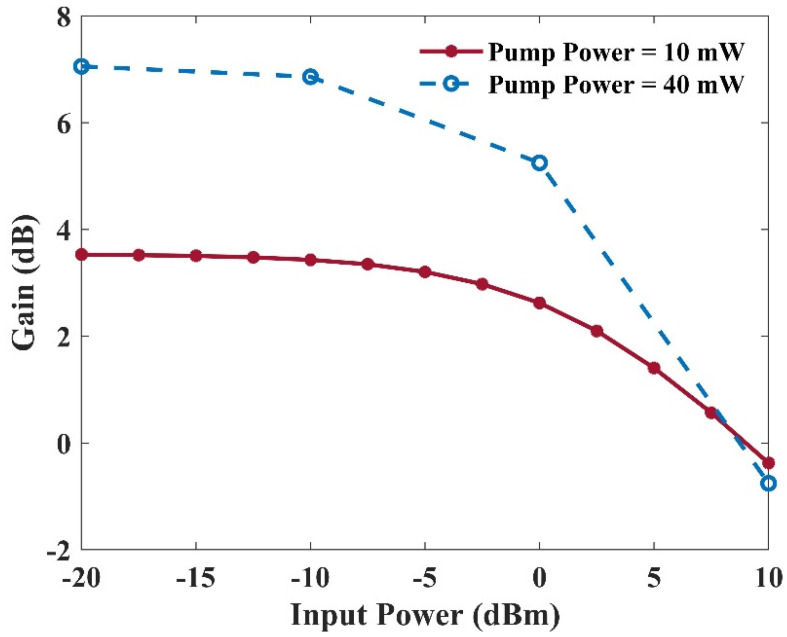
Gain versus input power with a 10 mW or a 40 mW optical pumping power.

**Table 1 nanomaterials-13-00685-t001:** Simulation parameters.

Symbol	Value	Description
*L*	2 mm	RSOA length
*H*	0.25 µm	Height of the RSOA
*W*	4 µm	Width of the RSOA
*α* _max_	1000 m^−1^	The maximum modal absorption coefficient
*α* _int_	200 m^−1^	Material absorption coefficient
*τ_w−r_*	1.4 ns	Recombination lifetime for WL
*τ_g−r_*	2.8 ns	Recombination lifetime for GS
*τ_e−r_*	2.8 ns	Recombination lifetime for ES
*τ_w−e_*	2 ps	Relaxation lifetime from WL to ES
*τ_e−w_*	1 ns	Escape lifetime from ES to WL
*τ_e−g_*	3 ps	Relaxation lifetime from ES to GS
*τ_g−e_*	20 ps	Escape lifetime from GS to ES
*D_e(g)_*	4 (2)	Degeneracy rate of ES (GS)
*n_r_*	3.5	Refractive index
*R*	0.8	Reflection of the mirror

**Table 2 nanomaterials-13-00685-t002:** Reported broadband optical amplifiers with their corresponding optical bandwidth and operation region.

Paper’s Title	Optical Bandwidth	Operation Regime
Wideband Gain MQW-SOA Modeling and Saturation Power Improvement in a Tri-Electrode Configuration [44]	76 nm	1470–1546 nm
A broad-band MQW semiconductor optical amplifier with high saturation output power and low noise figure [45]	120 nm	1450–1570 nm
C- and L-Band External Cavity Wavelength Tunable Lasers Utilizing a Wideband SOA With Coupled Quantum Well Active Layer [46]	77 nm	C-band and L-band
Ultrabroadband reflective semiconductor optical amplifier using superimposed quantum dots [16]	270 nm	460–730 nm
Switchable Ultra-wideband All-optical Quantum Dot Reflective Semiconductor Optical Amplifier	1000 nm	800–1800 nm

## Data Availability

Not applicable.

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
