# Peer review of "Switchable Ultra-Wideband All-Optical Quantum Dot Reflective Semiconductor Optical Amplifier"

_nanomaterials, 2023, doi:10.3390/nano13040685_

Round 1

Reviewer 1 Report

Manuscript reports on the theoretical analysis and calculations of properties of quantum-dot-based optical amplifier. The work touches actual topics of the development of optical devices. The approach looks promising. Nevertheless, I noticed several issues listed below, which could debase the research. I thus request major revision before the manuscript could be accepted in Nanomaterials.

1. line 100: Why the periodic array of QDs was considered? It does not look realistic in real structures.

2. The model defined by the set of equations (6)-(8) suffers from oversimplification neglecting important features that may significantly affect the device performance. Especially, authors should consider more than one electron (hole) populating single QD where moreover the number of electrons and holes may be different. Even if authors assume separated QDs, they may be charged through the wetting layer. If authors reckon such model extension too complex, they should at least extensively discuss relevant effects, including also Auger recombination, and estimate possible implications for their results. Also the spin of particles should be involved at the recombination dynamics.

3. line 205: Degeneracy rate should be defined.

4. Figure 16: Dimension unit of N _w should be completed.

5. The total occupation probability f+h in Figs. 15 and 17 is greater than 1. How can such results be interpreted?

Typing errors:

valance -> valence (4x)

line 191: Nw -> N_w

Author Response

Dear Editor

Enclosed is a revised version of our paper entitled “Switchable Ultra-wideband All-optical Quantum Dot Reflective Semiconductor Optical Amplifier” submitted for your consideration. We applied all comments in the body of the paper and highlighted them with red color and here for more simplicity, we write a short comment for each question in the following.

Bests

Ali Rostami

Reviewer 2 Report

The Manuscript describe about the Ultra wideband semiconductor. Below are my comments:

Add the flowchart after numerical algorithm section

R3 have higher energy!! What is the importance of having three different radius values made from InGaAs. Can’t we use uniform size material??

What is the multiple band in Figure 3, 4, 5 that lies between 1.2 to above 1.3. Mention the Y axis label.

What is the difference between Figure 2 and 7? Mention in the manuscript.

What is the significance of Interaction point. Mention in the manuscript.

Increase the font size of equations 6, 7, 8,10, 11, 12, 13

Provide the comparison table against the other proposed methods.

Author Response

(The authors gave the same response as above.)

Reviewer 3 Report

The authors have investigated a comprehensive study on quantum dot reflective semiconductor optical amplifiers with optical pumping in their reflective configuration. The dynamics of carrier densities, small-signal gain, and saturation attributes have been studied. The manuscript is interesting and useful for the design and application of quantum dot reflective semiconductor optical amplifiers. The paper is acceptable to be published in Nanomaterials, provided the following issue can be addressed

  1. Some abbreviations should be clarified when they appear for the first time.
  2. Add more details in the table to describe the physical parameters of the considered model.
  3. Add some discussion regarding the noise figure regarding the produced amplifier.
  4. Add some discussion either in the discussion Section or the Introduction regarding the potential application of the semiconductor amplifier in optical communications.

See e.g.

C Jin et al., Nonlinear coherent optical systems in the presence of equalization enhanced phase noise, Journal of Lightwave Technology, 2021.

H Schmeckebier et al., Quantum-dot semiconductor optical amplifiers for energy-efficient optical communication, Green photonics and electronics, 2017.

Author Response

(The authors gave the same response as above.)

Round 2

Reviewer 1 Report

Authors amended the manuscript according my recommendations. Though used approach remained so simple as before, presented results are sound enough to warrant the publication. I approve the publication of the manuscript in Nanomaterials.

Author Response

(The authors gave the same response as above.)

Reviewer 2 Report

Add your proposed methods optical bandwidth and operation regime values in Table 1. Place Table 1 before conclusion. Arrange the position of Table 1. 

Arrange the fonts size of Flowchart. Place in horizontal pattern for the text description. 

Add the Algorithms as Step1, Step 2 and so on. Revise the Numerical Algorithm section. 

Where in the manuscript you have described the Figure 10?

Check the flow sequence of Figures. Figure 10 is coming first and then again Figure 6 description is mentioned. 

I cannot understand your equation 9 format. Is it modulus values???

Line 238, 0.8 to 1.8??? 

Author Response

(The authors gave the same response as above.)

Round 3

Reviewer 2 Report

The authors seems to be rush. I would like to suggest that revised the manuscript as suggested. 

Do not place Table 1 in introduction section. Place it before conclusion part as suggested. 

In section 2.4, arrange each steps in separate lines. 

Revise line 187, Figure 10 shows the flowchart not algorithm.

The decision making stage of flowchart is not arranged properly. I would like to suggest to go through the flowchart presentation style. Do more study on flowchart presentation strategies. 

 As far as possible, maintain the flow sequence of Figure description. The figures should be described in particular flow. I have noticed that you have described suppose Figure 11, then again you describe Figure 2 and 6. 

Author Response

(The authors gave the same response as above.)

Round 4

Reviewer 2 Report

ok. Thanks for changes.